# The Need to Improve Riparian Forests Management in Uranium Mining Areas Based on Assessment of Heavy Metal and Uranium Contamination

**Václav Pecina** [1,2] , **David Juřička** [3,*], **Jindřich Kynický** [4], **Tivadar Baltazár** [2] ,
**Renata Komendová** [1] **and Martin Brtnický** [1,2,3]

1   Institute of Chemistry and Technology of Environmental Protection, Faculty of Chemistry,
    Brno University of Technology, Purkyňova 118, 612 00 Brno, Czech Republic; xcpecina@fch.vut.cz (V.P.);
    komendova@fch.vut.cz (R.K.); martin.brtnicky@mendelu.cz (M.B.)
2   Department of Agrochemistry, Soil Science, Microbiology and Plant Nutrition, Faculty of AgriSciences,
    Mendel University in Brno, Zemědělská 1, 613 00 Brno, Czech Republic; tivadar.baltazar@mendelu.cz
3   Department of Geology and Pedology, Faculty of Forestry and Wood Technology, Mendel University in Brno,
    Zemědělská 3, 613 00 Brno, Czech Republic
4   BIC Brno, Technology Innovation Transfer Chamber, Purkyňova 125, 612 00 Brno, Czech Republic;
    jindrak@email.cz
*   Correspondence: david.juricka@mendelu.cz; Tel.: +420-776-324-142

**Abstract:** Environmental contamination caused by uranium mining is becoming a worldwide issue due to its negative impact on the environment. The aim of this study is to evaluate the contamination levels of riparian forest stands and their interaction with pollutants on the example of two localities with long and short-term uranium mining closure. Notably high Cu content, which exceeded the lower range of the toxicity limit in 50–75% of the cases, was detected in the leaves. Increased U content also represents a potential risk. As both of the elements have a negative effect particularly on the root system, it can be assumed that the soil-stabilizing and water erosion-reducing functions of the stands may be reduced. Extremely high U content (51.8 mg/kg DA) in the leaves of *Aesculus hippocastanum* L. indicates its potential for phytoremediation. Significantly higher U content determined at the locality with the long-term closure of mining was probably caused by the instauration of the shallow hydrogeological circulation after mine inundation. Strong correlation between U and Pb suggests identical trend of their uptake and accumulation by plants. A significant dependence of the level of contamination on the distance from its source was not demonstrated. Therefore, the management of mining areas should focus on the protection of riparian forest, which can through its stabilizing and erosion-reducing functions and through suitable species composition effectively prevent spreading of contamination.

**Keywords:** bioaccumulation; riparian forest; uranium mining; *Aesculus hippocastanum*; phytoremediation; forest functions

## 1. Introduction

Environmental contamination in the aftermath of uranium mining has been an objective of immense scientific interest worldwide in recent decades [1–5]. Intensive U mining and inappropriate residual material management have resulted in negative environmental impacts [6]. Even though U occurs naturally in the environment [7], its elevated concentrations pose a serious risk to human health and all living organisms due to its toxicity and radioactivity [8–10]. Uranium in contaminated areas can easily enter the food chain through water and can also be absorbed from the soil by plants [4,7,9].

Although the bioavailability of U for vegetation is becoming the subject of a growing number of studies [2,11], the research issues of toxicity and mutagenicity of surface water and sediments in U mining areas have not received much attention so far [1].

Uranium mining is also associated with heavy metal contamination [5,6,12]. Heavy metals (e.g., As, Cd, Cu, Hg, Pb and Zn) are considered the most dangerous pollutant representatives owing to their toxicity, persistence and bioaccumulation [3,5], which is why they pose a permanent threat to the environment and humans [13]. Even though some heavy metals are essential for plants, they are all toxic in high concentrations [14]. It is, therefore, crucial to monitor U mining areas and, on that basis, adapt the management of such high-risk areas.

The history of U mining in the Czech Republic (former Czechoslovakia) is rich. After 1945 uranium became a strategic raw material for the military and the energy sectors and its importance increased significantly [15]. Out of the 164 investigated uranium deposits and occurrences in the Czech Republic 66 have been mined [16]. This publication focuses on significant localities with short and long-term U underground mining closure in the Rožná and Olší localities.

The Rožná deposit was discovered in 1954 [17]. The total area of the Rožná-Rozchody ore field is 1195 ha. The whole area had been heavily mined long-term, totaling more than 1000 km of excavated space. Uranium mining was conducted in the pits of Rožná 1–6 (RI–RVI), Bukov 1, Bukov 2 and Milasín in the scale of 400–450 t per year between 1962 and 1989. After 1989, mining declined due to the reassessment of U reserves in order to achieve greater economic efficiency. Selective mining commenced at the Rožná deposit in 1995 resulting in U production decrease to 300–350 t per year [18]. The main underground operating pit R1 was in continuous operation from 1957 to 31st December 2016. At the date of its closure it was the last functioning U mine in Central Europe.

Uranium mineralization at the Olší locality in the Olší-Drahonín deposit area was discovered during mineral exploration between 1954 and 1956. In 1958, the excavation of the Olší pit began, two years later the Drahonín pit was excavated. Mining reached its peak between 1966 and 1968, when the highest quality ore was mined, and since 1985 only residual reserves were mined. Mining was terminated in the first quarter of 1989, a total of 2,883,328 t of ore was mined, i.e., 2916.5 t of U [19].

Some terrestrial plants can be used for mining contamination biomonitoring due to their capacity to accumulate U and heavy metals [8]. However, it is essential to take into consideration the potential toxicity effects of pollutants on these plants, or rather entire forest stands, which are not scientifically addressed. The aim of the publication is to: (1) determine the content of U and heavy metals (Cd, Cu, Pb and Zn) in the leaves of the riparian forest stands of the U mining area with an emphasis on the most diverse species variability as a complex landscape element, not merely its fragment represented by one or two selected species; (2) evaluate the spatial distribution of contamination in the riparian forest stands in relation to the distance from the source of contamination; (3) evaluate the effectiveness of the measures adopted to eliminate the distribution of the contaminants; (4) assess the risks posed to local ecosystems and their current management.

## 2. Materials and Methods

### 2.1. Study Area

The research subjects of interest are the Nedvědička stream in the mining area Rožná-Rozchody (site 1) and the Hadůvka stream in the mining area Olší-Drahonín (site 2) (Figure 1). The streams are the main recipients of contamination by radionuclides and other pollutants. Radionuclides are introduced to Nedvědička stream via the associated wastewater discharge point, where waters from the mine water decontamination station, the sludge treatment plant, and the active sewage treatment plant of the area chemical treatment plant mix. Water from the mine water treatment (MWT) of the Olší-Drahonín water catchments flows into Hadůvka stream.

The watercourses create alluvial floodplains of a very limited extent. The average annual flow of Nedvědička and Hadůvka stream are 0.212 m$^3$/s [20] and 0.019 m$^3$/s [21] respectively. Riparian

forest stands of the watercourses consist of fast growing and ameliorative species related to higher groundwater saturation (especially *Alnus glutinosa* L. and *Acer platanoides* L.).

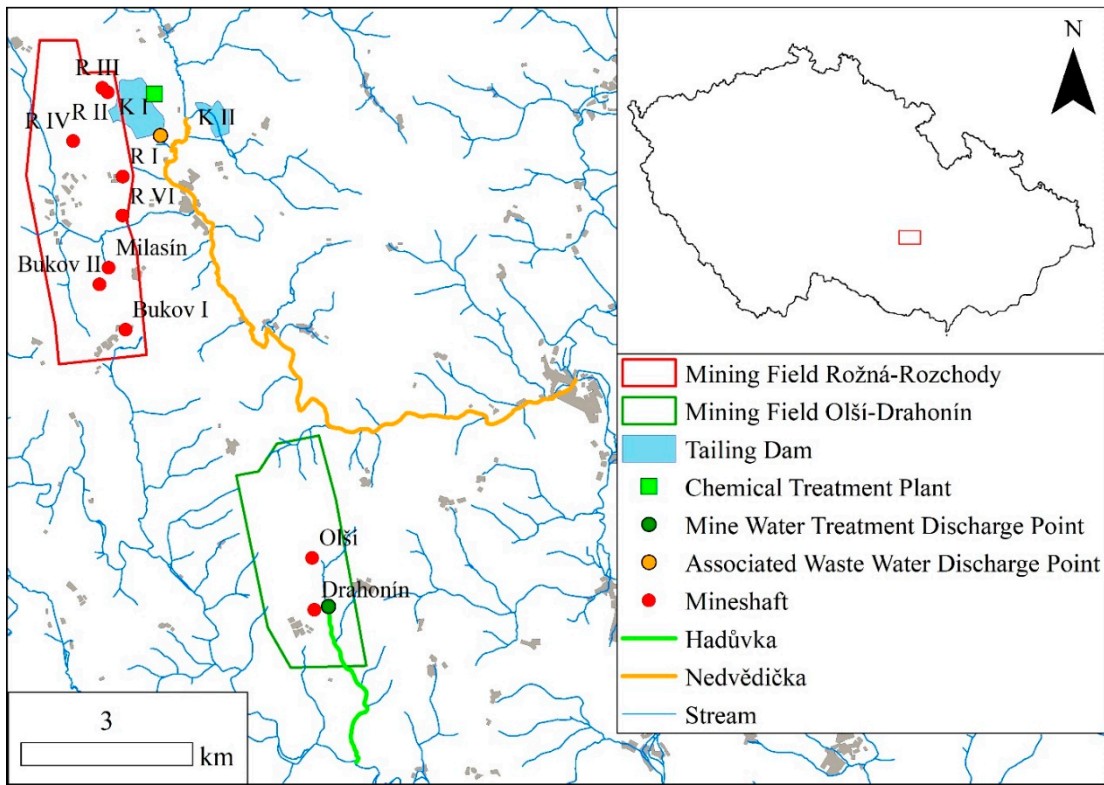

**Figure 1.** The area of interest.

The recipient waters of the Hadůvka and Nedvědička streams are monitored for radionuclides and other pollutants annually. According to CHMI [22], the Nedvědička stream on the Dvořiště and Nedvědice profiles fell into the category of very heavily polluted water in the 2017–2018 period according to the surface water quality classes (WQC) in compliance with the ČSN 75 7221. WQC limit values for this category are ≥ 1000 and ≥ 2000 mBq/L for gross α and gross β. Total U concentrations exceeded 200 μg/L.

The Hadůvka stream is monitored on the Skryje profile. PMO [23] states the same categorization for the 2017–2018 period for this profile as in the case of Nedvědička stream for gross α, gross β and U. Gross α 12,460 mBq/L, gross β 2590 mBq/L and U concentration 338 μg/L were measured on the profile in 2017. Hudcová et al. [1] state the average value of $U_{nat}$ to be 117 μg/L on the Skryje profile, which exceeds the USEPA [24] maximum concentration level of U in drinking water (30 μg/L $U_{nat}$).

The heavy metal and radionuclide contents of the Hadůvka locality sediments were reported by Hudcová et al. [1]. The $U_{nat}$ concentrations on the Olší, MWT and Skryje profiles were measured as 182, <5 and 44.3 mg/kg of dry weight (DW) respectively, Cd 1.03, 1.14 and 0.710 mg/kg DW, Pb 38.5, 34.5 and 28.2 mg/kg DW, Cu 40.1, 79.0 and 30.1 mg/kg DW and Zn 203, 297 and 149 mg/kg DW. The $^{235}U$ isotope activity in the sediments was below the detection limit [22].

Hence, these are heavily polluted localities where the mining activity has endangered the environment by surface water and groundwater contamination [1].

### 2.2. Sampling Design

The selection of the riparian forest areas for research was based on the study by Juřička et al. [25]. The design of sampling took into account the morphology of the watercourses and the distance from the point source of pollution. Selected sampling sites allowed for sedimentation and accumulation of

fine, potentially contaminated sludge, which is introduced into the watercourses from the point sources of pollution. When identified and verified, final selection of the distinct sludge sedimentation areas was carried out so that the individual sampling points covered the upper, middle, and lower parts of the watercourse, where plot 1 was the closest to the pollutant emissions area (contamination hotspot).

Samples were collected from 10 sampling plots at both sites. Monitored watercourse sections (distance between the first and last sampling) measured 13.5 km at site 1 (443–365 m a.s.l.) and 4 km at site 2 (412–367 m a.s.l.). Tree species composition was surveyed at all the studied plots. Two samples from two different representative tree individuals were taken at each sampling plot.

### 2.3. Sample Collection and Preparation

Leaves of adult tree individuals were sampled at a 2-meter height above the ground at different tree sides in the autumn aspect (October). The sampled trees were up to 1.5 m away from the watercourse. All the leaves were sampled at the same time period. Due to the different species composition, it was not possible to sample by dormancy. All represented deciduous tree species of the riparian forest stands at both sites were sampled.

The samples were placed into paper bags and immediately transported to the laboratory in a cooling box (5 °C) for further analyses. In the laboratory, the samples were washed in distilled water and dried to constant weight at 65 °C. Subsequent dried sample preparation consisted of grinding into powder and homogenizing.

### 2.4. Sample Analysis

Chemical analyses were performed from dry ash (DA). DA is noted as the best source material for evaluating the pollution status of forests, especially in large-scale surveys [26]. The advantages are: (1) heavy metal contents in DA of leaves are generally more homogeneous and better correlated with corresponding values in soil than those in dry weight (DW); (2) heavy metal content in DA of leaves is less dependent on sampling methodology; and (3) DA is a better reference base to evaluate pollutant contents in litter and thus preferred in forest management and protection of soils [26–28]. DA to DW ratio of leaves was determined after ashing [27] for result comparison with similar studies, because DW contents are usually used as a reference base.

High purity leaf ash of 40 representative leaf samples was obtained through carbonization/digestion under 200 °C. Ash was homogenized and sampled with the weight of 1 g per sample. A standard cluster method for effective homogenization was performed. Digestion of samples was accomplished according to EPA methods: EPA 3050B [29] and EPA 200.8 [30]. To accomplish complete decomposition, the samples were boiled in 15 mL of nitric acid p.p. (65%) with continuous additions of hydrogen peroxide (3 mL), filtered, filter boiled with the addition of 10 mL of aqua regia (3 HCl:1 $HNO_3$), filtered, filter boiled again with the addition of 10 mL of aqua regia (3 HCl:1 $HNO_3$) and filtered. All 3 solutions were mixed and topped up to 100 mL with demineralized water. Blanks were prepared by identical procedure with identical amounts of acids and hydrogen peroxide, only without the sample. The samples were filtered to 50 mL volumetric flasks and measured after cooling.

A Spectro Ciros CCD ICP-OES (Spectro Analytical Instruments, Lithea s.r.o., Brno, Czech Republic) with axial plasma was applied to determine 5 elements from the high purity leaf ash. A 20–23 short depth-of-field lens system allowing a more uniform region of the plasma to be viewed was used. Gas flows were maintained at the same rate for all the analyses. The 214 nm, 324 nm, 168 nm, 367 nm, and 213 nm lines were used for Cd, Cu, Pb, U, and Zn determination. Further instrument settings: carrier gas was argon, oscillator power 1600 W, Mod-Lichte nebulizer. Standard repeatability of the method, including the extraction procedure, was assessed with the use of the standard samples. The duplicates of one NIST standard material were prepared per batch and the results compared to the consensus values.

Water used in the experiment was demineralized twice by reverse osmosis and by two ion exchange columns with mixed-bed with resulting conductivity 0.07–0.1 μS/cm. Nitric acid (Penta,

Czech Republic) was of purity for semiconductors, hydrochloric acid (Penta, Czech Republic) was of purity for analysis. A referenced blank was established for all measurements. Standards were prepared by dissolving pure elements except for U, which was prepared from overdried pure Merck $UO_3$ oxide dissolved in $HNO_3$. Limit of detection (LOD) of used nebulizer was set to be 10× more accurate than usual with approximate 0.X ppb of analyte in solution. Limit of quantification (LOQ) was approximately 3× higher.

## 2.5. Data Analysis

Range of values by Kabata-Pendias [7] was used to evaluate the level of heavy metal contamination in trees. The range values are 5–30, 20–100, 30–300 and 100–400 (all mg/kg) for Cd, Cu, Pb and Zn respectively and are based on a large scale review of plant sensitivity values that are excessive or toxic and do not include highly sensitive or highly tolerant species.

Data processing and statistical analyses were carried out in the freely available R program version 3.6.3. Ref. [31] together with RStudio [32]. Advanced graphs were created with the use of additional packages "ggplot2" [33] and "dplyr" [34].

Relation between the heavy metal and U content in leaves of different trees with species, site and plot dependence, was characterized by one-way analysis of variance (ANOVA) type I (sequential). Sum of squares was used at the significance level of 0.05 [35], where the heavy metal and U content was applied as a continuous response variable and species, site or plot as categorical explanatory variable. Partial eta-squared ($\eta p^2$) from the "BaylorEdPsych" package [36] was used to measure the effect size. Dunnett-Tukey-Kramer pairwise multiple comparison test from the "agricolae" package [37] was applied to determine statistically significant difference among factor level means. "Treatment contrast" was used to calculate factor level means with 95% confidence interval [38]. Pearson's correlation coefficient was used to establish the statistical relation between the heavy metal and U contents.

All statistical models were post-analysis checked at the significance level set to 0.05. Kolmogorov–Smirnov and Shapiro-Wilk normality tests from the "nortest" package [39] were used to test the normality and the Bartlett's and Levene's tests from the "car" package were applied [40] to test the homoscedasticity. The models were repeatedly checked with the help of different diagnostic plots [41]: plot of residuals versus fitted values, normal Q-Q plot of standardized residuals, scale-location plot, plot of residuals versus leverage, Cook's distance plot and plot of Cook's distance versus leverage.

## 3. Results and Discussion

### 3.1. Evaluation of Heavy Metals and Uranium in Vegetation

A result summary of the heavy metal and U contents in the tree leaves is in Table 1. The results show considerable variability of the target element contents in the riparian forest stands influenced by the sampling site, but also significantly by tree species. Based on the one-way ANOVA results, a statistically significant species dependent difference between the contaminant contents in the leaves is apparent in the cases of Cd ($F_{15,24} = 4.21$, $p < 0.001$, $\eta_p^2 = 0.72$), Pb (F15,24 = 241.84, $p < 0.001$, $\eta_p^2 = 0.99$) and U ($F_{15,24} = 7.27$, $p < 0.001$, $\eta_p^2 = 0.82$). No statistically significant difference was found in the case of Cu ($F_{15,24} = 0.59$, $p = 0.86$, $\eta_p^2 = 0.27$) and Zn ($F_{15,24} = 1.19$, $p = 0.34$, $\eta_p^2 = 0.43$). Significant variations in target element accumulation among plants are also mentioned, for instance, by Kabata–Pendias [7], or Li et al. [42].

**Table 1.** Heavy Metal and Uranium Content (mg/kg DA) in the Leaves of the Riparian Forest Stands.

| Plot | Nedvědička (Site 1) | | | | | | Hadůvka (Site 2) | | | | | |
|---|---|---|---|---|---|---|---|---|---|---|---|---|
| | Tree Species | Cd | Cu | Pb | U | Zn | Tree Species | Cd | Cu | Pb | U | Zn |
| 1 | *Alnus glutinosa* L. | 0.17 | 21.9 | 7.36 | 0.30 | 108 | *Aesculus hippocastanum* L. | 0.73 | 24.7 | 326 | 51.8 | 38.6 |
| | *Salix fragilis* L. | 0.59 | 23.3 | 5.90 | 3.60 | 63.4 | *Corylus avellana* L. | 0.78 | 118 | 11.6 | 10.5 | 86.6 |
| 2 | *Alnus glutinosa* | 0.31 | 22.5 | 6.41 | <0.1 | 59.6 | *Fagus sylvatica* L. | 0.38 | 48.3 | 15.4 | 10.1 | 57.0 |
| | *Alnus glutinosa* | 0.30 | 12.8 | 4.28 | <0.1 | 96.2 | *Acer platanoides* L. | 0.23 | 35.6 | 8.81 | 1.81 | 78.8 |
| 3 | *Cornus sanguinea* L. | 0.15 | 10.9 | 2.59 | 2.71 | 34.1 | *Carpinus betulus* L. | 0.50 | 24.2 | 15.4 | 19.7 | 62.6 |
| | *Alnus glutinosa* | 0.27 | 34.2 | 1.84 | 3.51 | 101 | *Crataegus oxyacantha* L. | 0.19 | 9.49 | 5.21 | 3.88 | 52.1 |
| 4 | *Acer platanoides* | 0.74 | 16.9 | 3.78 | <0.1 | 79.1 | *Sambucus nigra* L. | 0.20 | 16.0 | 6.80 | 1.99 | 54.6 |
| | *Salix fragilis* | 2.02 | 21.9 | 4.15 | <0.1 | 64.8 | *Fagus sylvatica* | 0.24 | 18.1 | 13.2 | 1.15 | 75.7 |
| 5 | *Salix fragilis* | 1.07 | 16.7 | 4.54 | <0.1 | 106 | *Alnus glutinosa* | 0.18 | 16.5 | 9.31 | 9.80 | 57.1 |
| | *Acer negundo* L. | 0.13 | 20.7 | 3.24 | <0.1 | 40.9 | *Alnus glutinosa* | 0.64 | 29.8 | 28.1 | 16.4 | 66.7 |
| 6 | *Alnus glutinosa* | 0.12 | 30.8 | 4.73 | 5.73 | 79.5 | *Sorbus aucuparia* L. | 0.23 | 13.4 | 6.13 | 9.12 | 34.4 |
| | *Salix fragilis* | 2.04 | 13.9 | 2.87 | <0.1 | 70.6 | *Quercus robur* L. | 0.24 | 20.4 | 3.84 | 10.9 | 54.0 |
| 7 | *Alnus glutinosa* | 0.19 | 23.2 | 6.63 | <0.1 | 103 | *Acer platanoides* | 0.71 | 27.1 | 3.25 | 3.15 | 111 |
| | *Acer platanoides* | 0.53 | 11.6 | 3.52 | <0.1 | 81.9 | *Acer platanoides* | 0.49 | 69.6 | 4.12 | 0.42 | 58.3 |
| 8 | *Alnus glutinosa* | 0.12 | 19.0 | 2.12 | <0.1 | 68.0 | *Fraxinus excelsior* L. | 0.23 | 36.6 | 6.68 | 0.24 | 75.1 |
| | *Salix fragilis* | 2.14 | 23.0 | 3.24 | 2.71 | 109 | *Corylus avellana* | 0.88 | 22.1 | 9.47 | 1.66 | 144 |
| 9 | *Acer pseudoplatanus* L. | 0.16 | 24.8 | 3.57 | 7.36 | 48.2 | *Corylus avellana* | 0.25 | 32.9 | 15.1 | 0.38 | 67.4 |
| | *Sambucus nigra* | 0.22 | 14.1 | 3.32 | <0.1 | 62.5 | *Carpinus betulus* | 0.21 | 20.3 | 5.89 | 1.16 | 35.7 |
| 10 | *Corylus avellana* | 0.16 | 17.7 | 4.53 | 1.50 | 35.3 | *Alnus glutinosa* | 0.45 | 27.6 | 9.72 | 2.81 | 110 |
| | *Alnus glutinosa* | 0.24 | 17.9 | 5.17 | 4.82 | 60.8 | *Prunus avium* L. | 0.23 | 30.8 | 8.56 | 2.82 | 49.7 |

When attention is aimed at heavy metals, it is evident that the content of Cd and Pb in the leaves at site 1 did not exceed the lower toxicity limit in any case. The Cu content, which exceeded the lower range of the toxicity limit in 50% of the cases, could be classified as potentially problematic and the lower limit of Zn was exceeded in 25% of the cases. The state of affairs of Cd at site 2 is equivalent to site 1. The situation is also similar in case of Cu, although in this time the lower range of the toxicity limit was exceeded in almost 75% of the cases. The lower toxicity limit was exceeded once in Pb and twice in Zn. It is, therefore, possible to presume manifestations of toxicity associated mainly with Cu in case of trees with higher sensitivity. Although Cu is an essential element for plants [43] it can be toxic in higher concentrations [7]. Copper is more phytotoxic compared to other potentially toxic elements, such as Cd and Zn [44]. It inhibits many enzymes, photosynthesis, nitrogen fixation and respiration and thus also negatively influences the vegetative growth [7,45] and its rhizotoxicity is of exceptional importance [44].

Similar accumulation trend is visible, especially in the case of Zn, when the most commonly occurring species (*n* > 5) of the stands are compared (Figure 2). The highest Cu content was identified in the *Acer platanoides* leaves, whereas Pb and U were contained in the *Alnus glutinosa* leaves the most. Nevertheless, out of the less common species *Corylus avellana* L.leaves reached the highest average Cu and Zn values of 47.8 mg/kg DA and 83.1 mg/kg DA respectively, suggesting its possible use for biomonitoring. Biomonitoring studies on *Corylus* spp. are not numerous [46], however, the results of other authors also suggest that *C. avellana* shows a promising application [46,47].

High U value variability unrelated to pollution source distance, plots and individual species (Table 1, Figure 2) indicates a significant influence of environmental factors and, in particular, of varied accumulation abilities of individual species. Malaviya and Singh [11] or Favas et al. [8] reached similar conclusions.

Information available on U toxicity in plants grown in U-contaminated soils is limited and often contradictory [11,48,49]. Even though a small amount of U in the soil can increase the root and shoot production [48,49], high values measured in this area [1,22,23] may be toxic. Soil U seriously affects photosynthesis, inhibits plant growth, reduces seed germination and plant survival capacity [9,48,49]. Its toxicity varies among species [7,9], and so different manifestations may occur depending on species composition of the assessed riparian stands. Further research is needed in this field.

Uranium phytotoxicity of riparian forests may be further enhanced by the nature of soil. For instance, pseudogley soils, which are more common in such conditions, evince higher U-toxicity

values than, for example, chernozems. It is due to their more acidic nature and predominantly reductive (anoxic) environment, resulting in higher mobility, bioavailability [48] and U speciation favoring high free uranyl ($UO_2^{2+}$) concentration, which is expected to be a key toxic U species [50,51].

In order to compare the results to other tree species from U-contaminated areas recalculation to DW was needed. Favas et al. [8] state, for instance, almost twice as high values (1.2 mg/kg DW) of willow species (*Salix atrocinerea* Brot.) as willow species (*Salix fragilis*) assessed in this study (0.76 mg/kg DW). Hinton et al. [52] mention the potential of *Nyssa sylvatica* Marshall and *Liquidambar styraciflua* L. to remove U from contaminated soil with leaf U contents of 5.4 mg/kg DW and 8.2 mg/kg DW respectively.

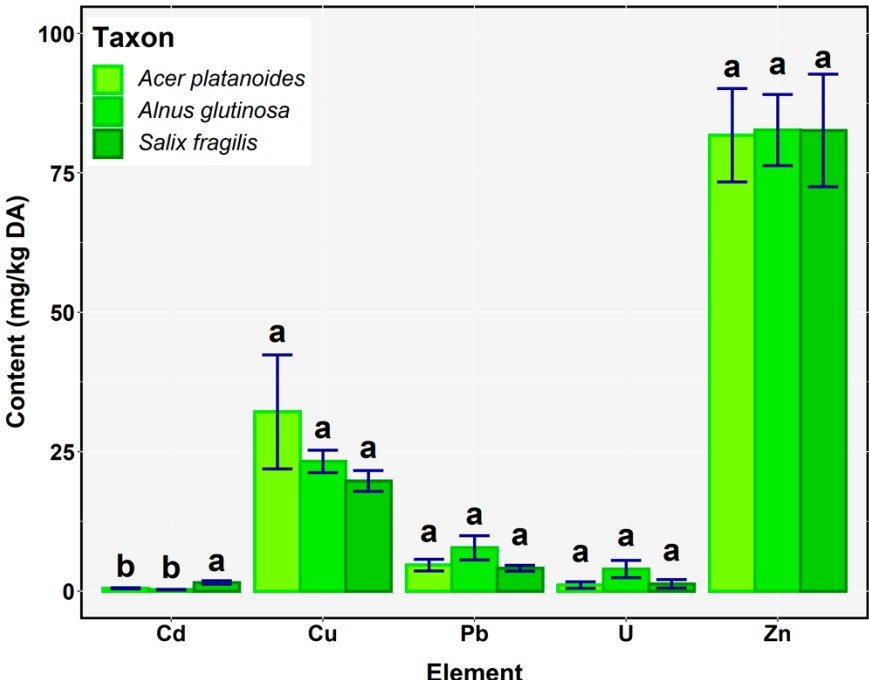

**Figure 2.** Heavy metal and U content comparison in the leaves of the dominant tree species. Different letters represent statistically significant differences at $p < 0.05$.

High U content (17.0 mg/kg DW) was found out in the leaves of *Aesculus hippocastanum*. This anomaly has two possible explanations. The sediments and the water in this area were either heavily contaminated by the inflow of contaminated seepage water from the Drahonín mine dump, or the *A. hippocastanum* species shows higher potential than other monitored tree species to accumulate U. Given that the observed content of U in *C. avellana*, growing in the same area was many times lower may indeed indicate the potential of *A. hippocastanum* to accumulate U and thus be crucial in phytoremediation of U-contaminated soils. According to the categorization presented by Favas et al. [8], *A. hippocastanum* falls into the category of high accumulators (range of mean U concentration = 10–100 mg/kg). However, further research is needed in this respect. On the other hand, high U content in leaves under inadequate management can result in easy further environmental spread through the food chain.

It can be assumed that the total U content in the assessed tree species is even higher due to the fact that the plant root U content is significantly higher than that of their above ground biomass [2,8,10,11]. Despite the fact that U and heavy metals are released into the environment in harmless concentrations, owing to their accumulation in recipients during the pollution source operation, their concentration can reach risk values over the long term. Although the phytostabilization function of riparian forests is important, with the increasing accumulation of pollutants, the risk of their spread to other components of the ecosystem through the food chain also increases with inadequate management of polluted areas.

Potential synergic negative effect of Cu rhizotoxicity [44] and U toxicity on the root systems of trees presents the main danger for the studied riparian forests. For example, Stojanović et al. [48] report 54.9% mass reduction of roots in highly U-contaminated soil (1000 mg/kg). Similarly, Hou et al. [49] state that soil U content increase leads to the growth inhibition and root system development, because plant roots are sensitive to U stress. Such significant negative effects on the root system can reduce the most important riparian forest role of soil-stabilization and the function of water erosion mitigation.

### 3.2. Comparison of Pollution at the Sites of the Short-Term and Long-Term Closure of Mining

According to the site comparison in Figure 3, there is no statistically significant difference in the Cd content between the sites ($F_{1,38} = 1.30$, $p = 0.26$, $\eta_p^2 = 0.03$). The average Cd content was 0.58 mg/kg DA at site 1 and 0.40 mg/kg DA at site 2. However, there is significant difference in the Cu content between the sites ($F_{1,38} = 4.73$, $p = 0.04$, $\eta_p^2 = 0.11$). The average Pb content at the site 2 (25.6 mg/kg DA) is 6x higher than at site 1 (4.19 mg/kg DA), nevertheless, there is no statistically significant difference between the sites ($F_{1,38} = 1.83$, $p = 0.18$, $\eta_p^2 = 0.05$). The reason for the large difference between the averages is caused by an extreme value measured at site 2 (326 mg/kg DA). The statistically significant difference in the U content between the sites ($F_{1,38} = 5.56$, $p = 0.02$, $\eta_p^2 = 0.13$) was observed there. The average U content was 1.67 mg/kg DA at site 1 and 7.98 mg/kg DA at site 2. Zn exhibits no statistically significant difference between the sites ($F_{1,38} = 0.39$, $p = 0.53$, $\eta_p^2 = 0.01$).

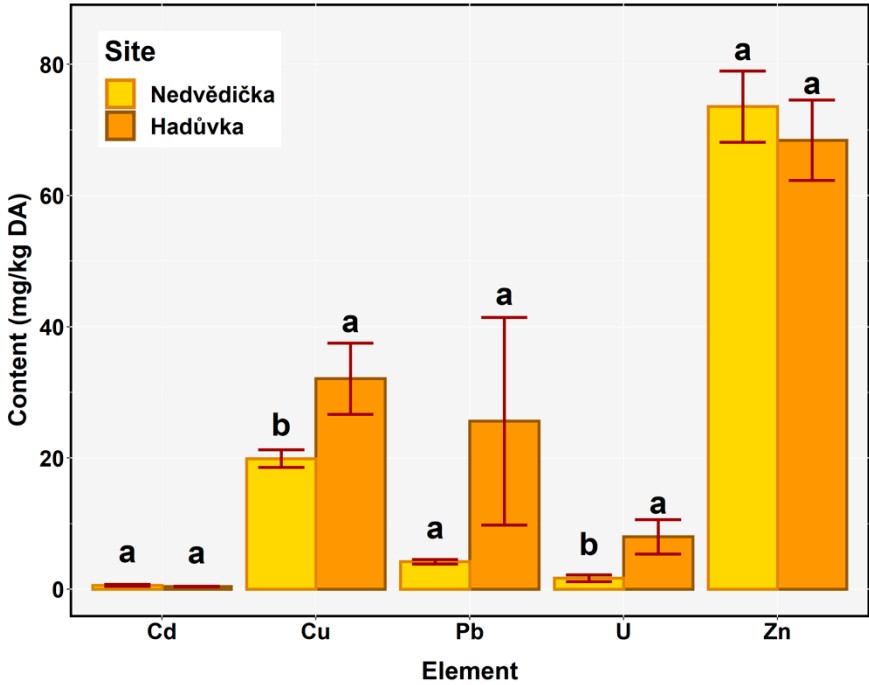

**Figure 3.** The comparison of the average heavy metal and uranium contents between the sites. Different letters represent statistically significant differences at $p < 0.05$.

It can be assumed that the statistically significantly higher U contents at site 2 are related to the closure of the mine in 1989 and its inundation. The groundwater level in the vicinity of the mine has been gradually increasing, the shallow hydrogeological circulation was restored followed by the Hadůvka stream water level increase due to the mine inundation. Even though the overbalanced mine water is continuously pumped and treated at the decontamination station, the instauration of the shallow groundwater circulation results in the contaminated groundwater penetration into the ecosystem and the area of interest surroundings. Říčka [53] describes the situation and mentions highly U-contaminated water springs (2.2 mg/L) in the vicinity of plot 10. Contrarily, the Rožná mine at site 1 was closed in 2016, removal work is still in progress and groundwater is pumped and treated at the

MWT plant in the MWT complex. Mine inundation to the highest planned level, i.e., the level where the instauration of shallow hydrogeological circulation can be expected, may occur no earlier than 8 years after the beginning of mine inundation [54]. At present, it cannot be assumed that the hydrogeological circulation and the entry of contaminated mine water into the ecosystem will be restored. However, lower values of U and heavy metals detected in the tree leaves at the site 2 support the theory.

### 3.3. Distribution of Contamination Depending on the Distance from the Source of Pollution

Although ANOVA showed no or borderline statistically significant difference (probably due to the lack of the sample) between the heavy metal contents in the leaves based on the distance from the contamination hotspot, treatment contrast already confirmed statistically decreased Cd, Cu, Pb and U contents in the leaves further from the contamination hotspot (Table 2). However, this decrease is not linear in all the cases.

**Table 2.** *p*-values of the heavy metal and uranium contents in the leaves (DA) with dependence of distance from the contamination hotspot (statistically significant differences from the plot 1 are highlighted in bold).

| Plot | Nedvědička (Site 1) | | | | | Hadůvka (Site 2) | | | | |
|---|---|---|---|---|---|---|---|---|---|---|
| | Cd | Cu | Pb | U | Zn | Cd | Cu | Pb | U | Zn |
| 2 | 0.92 | 0.54 | 0.27 | 0.46 | 0.78 | 0.05 | 0.25 | 0.05 | **0.03 *** | 0.85 |
| 3 | 0.82 | 1.0 | **0.002 **** | 0.64 | 0.53 | 0.08 | **0.046 *** | **0.048 *** | 0.09 | 0.86 |
| 4 | 0.20 | 0.69 | **0.03 *** | 0.46 | 0.63 | **0.03 *** | **0.046 *** | **0.048 *** | **0.02 *** | 0.93 |
| 5 | 0.77 | 0.63 | **0.03 *** | 0.46 | 0.66 | 0.13 | 0.07 | 0.06 | 0.11 | 0.98 |
| 6 | 0.36 | 0.98 | **0.03 *** | 0.69 | 0.70 | **0.03 *** | **0.046 *** | **0.04 *** | 0.07 | 0.53 |
| 7 | 0.98 | 0.52 | 0.18 | 0.46 | 0.82 | 0.47 | 0.36 | **0.04 *** | **0.02 *** | 0.46 |
| 8 | 0.33 | 0.84 | **0.004 **** | 0.82 | 0.93 | 0.36 | 0.11 | **0.046 *** | **0.01 *** | 0.13 |
| 9 | 0.80 | 0.69 | **0.02 *** | 0.47 | 0.30 | **0.03 *** | 0.09 | **0.049 *** | **0.01 *** | 0.70 |
| 10 | 0.81 | 0.56 | 0.13 | 0.62 | 0.20 | 0.07 | 0.11 | **0.047 *** | **0.02 *** | 0.56 |

* statistically significant difference at 5% significance level; ** statistically significant difference at 1% significance level.

Although there is usually a significant decrease in water and sediment pollution with distance from the source of pollution [12,13], the decrease was not conclusive in this case. The absence of a demonstrably significant dependence of the environmental contamination level on the distance from the source of pollution means that the monitored section lengths of the streams of site 1 (13.5 km) and site 2 (4 km) at flow rates of 0.212 and 0.019 $m^3$/s, are not sufficiently long for such dilution or capture of occurring pollutants that would significantly reduce the contamination of the riparian forest stands.

Nonetheless, other feasible factors may exist. For example, Faměra et al. [55] also mention the fact that pollutant concentrations did not decrease with increasing distance from pollution sources. As they observed the highest pollution levels in the furthest sampled place represented by a reservoir filled with sediment during the maximum pollution period, they inferred the watercourse morphology to be responsible. Although in this case the sites that allowed the sedimentation of fine, potentially contaminated sludge were methodically selected in the same way, each site is unique in its morphology, so that different accumulations could occur, which subsequently manifested itself in different values in tree species.

Besides, the findings can be closely and most likely related to the riparian ecosystem tree properties, as the specific bioaccumulation capacity of individual tree species [7,8] causes pollutants to be effectively captured in distant sections of the watercourse (bioaccumulation aspect). Additional argument may refer to a random distribution of species resistant to the toxicity (toxicant resistance aspect) with uninhibited and well-developed root system able to effectively capture contamination. This poses an increased risk due to the distribution of pollutants through the substance turnover to other components of the environment on a regional scale. In this context, proper ecosystem management of pollution recipients is very important.

### 3.4. Correlation Analysis

When determining the relation between individual elements (Table 3), particularly a strong correlation between U and Pb was found among different tree species indicating the same accumulation trend of these elements in their leaves. The relation between U and Pb in U mining areas was observed in the past, but with ambiguous outcomes. The study of Bai et al. [6] of topsoil contamination by radioactive materials and heavy metals identified increased Pb content; however, there was no statistically significant correlation between U and Pb. Similar research study by Haribala et al. [3] mentions increased Pb content without a statistically significant correlation between U and Pb. Similar situation is also described by Neiva et al. [4]. On the contrary, Wang et al. [5] reached opposite conclusions when focusing on the soil, water, and rice contamination. This emphasizes the link between Pb and U, together with the fact that their background content was higher than in other mines, which may indicate their common natural origin. In the study of plants growing on U mining waste, Li et al. [42] present the potential of *Phragmites australis* Cav. for phytoremediation of U and Pb contaminated soils. Similarly, Duquène et al. [56] report a joint increase in U and Pb uptake in shoots of *Brassica juncea* L., which may indicate a similar course of their uptake. Based on the results of this particular and other studies, it can be concluded that despite the different potential of plant species to accumulate target elements [7,41], the accumulation U-Pb trend may occur in relation with uranium mining-disturbed ecosystems and with a wide range of tree species. Since research suggests U-Pb association in the soil of U mining areas, this fact can be used in phytoremediation of these areas. However, their relation may also be generally associated with natural environmental factors, such as the background content of soil elements and, therefore, it is good to focus on the research into their associations in the future.

**Table 3.** Correlation matrix of Heavy Metals and Uranium.

| **Cd** | 0.042 | 0.067 | 0.018 | **0.32 *** |
|---|---|---|---|---|
| | **Cu** | 0.015 | 0.12 | 0.12 |
| | | **Pb** | **0.88 *** | −0.20 |
| | | | **U** | −0.28 |
| | | | | **Zn** |

* statistically significant difference at 5% significance level; *** statistically significant difference at 0.1% significance level.

### 3.5. Measures against the Spread of Contamination in the Environment

Although the issues of toxicity and mutagenicity of surface water and sediments in U-mining areas have not received much attention so far [1], this topic is becoming a growing subject of interest, especially in the context of soil contamination [3,4,6,10]. In connection with this problem, measures against the spread of contamination to the environment and the remediation of contaminated soils were also discussed.

The state enterprise DIAMO implements measures limiting the spread of radionuclides to the environment of the investigated area, which are generally recommended in the literature and previous research [13]. In particular, it involves thorough treatment of all types of water contaminated with radionuclides released into the environment and the reduction of sludge dust. It also operates a monitoring network to assess water and air quality or radiation load around access roads in the mining area and MWT.

However, these measures represent a response to newly emerged undesirable situations and fail to be fully effective due to locally elevated U and heavy metal values (Sections 2.1 and 3.1). When managing areas where radionuclides or metals are emitted into the environment, neighboring ecosystems should also be taken into account, as they may be another important factor contributing to limiting the spread of contamination further into the environment. In particular, the application of phytoremediative plants should be considered [2,10,11,42,52]. However, it is important to use not

only phytoextraction or phytostabilization of one particular species, but also to work at the ecosystem level. Appropriate forest ecosystem management can for instance significantly slow down the spread of contamination in the environment by the appropriate species composition with a predominance of species that do not acidify the environment [25]. Similarly, Sha et al. [10] state that the use of artificially constructed plant community plots can lead to the enhancement of U-contaminated soil phytoremediation.

In the management of such areas, especially in forestry, it is necessary to give priority to species that are resistant to contamination and at the same time contribute to reducing the spread of contamination. In such areas, the function of trees should be specifically protective, unless an intensive management associated with the collection (harvesting) of contaminated phytomass is in place, where it is safer to use phytostabilisation species or species accumulating U and heavy metals only in the roots, thus reducing the risk of further spread of toxic substances into the food chain. Riparian forest management is critical to soil-stabilizing and water erosion-reducing functions and can effectively prevent the transport of pollution to more distant areas. With the right choice of tree species composition, the spread of environmental contamination can be limited at both geographical and landscape levels.

## 4. Conclusions

Significant variability in the content of Cd, Cu, Pb, U and Zn was found in the leaves of riparian forest stands in the area of U mining. The variability was affected by the site, but also significantly by tree species. In particular, the increased Cu content, which exceeded the lower range of the toxicity limit in 50–75% of the cases, poses a potential risk that may result in negative effects on the prosperity and function of the riparian forest stands. Detected U contents may be toxic and since both Cu and U have a largely negative effect especially on the root system, it can be assumed that the soil-stabilizing and water erosion-reducing functions of the riparian forest stands may be reduced.

*C. avellana* shows a potential to be used in biomonitoring, particularly in the case of Cu and Zn contamination. High U content (51.8 mg/kg DA or 17.0 mg/kg DW) ascertained in the leaves of *A. hippocastanum* indicates the potential of the species for increased U accumulation and thus for phytoremediation of U-contaminated areas.

Despite the fact that no statistically significant difference between the localities with the short-term and long-term closure of mining was found in the case of the heavy metals (except Cu), variability was demonstrated in the case of U. Higher U values in the riparian forest stands of the long-closed mining site are probably related to inundation of the old mine, which led to the instauration of shallow hydrogeological circulation followed by contaminated groundwater penetration into the Hadůvka stream system.

No significant dependence of the riparian forest contamination level on the distance from the pollution source was found which can probably be contributed to a combination of factors: insufficient length of the investigated sections, in which the dilution factor was not sufficiently monitored; different accumulation in sediments as target element sources for trees on the basis of specific morphology of the watercourse; and the specific bioaccumulation capacity and toxicant resistance of individual tree species, which caused pollutants to be more efficiently captured merely in more distant sections of the watercourse.

Strong correlation between U and Pb suggests equivalent accumulation trend of these elements in the leaves of the riparian forest stands despite their different species composition. U and Pb correlations may indicate similar plant uptake trends. Since other studies suggest U-Pb association in the soil of U mining areas, the fact can be applied in phytoremediation of these areas.

Detected contamination levels show that the implementation of measures preventing the spread of contamination recommended by previous research is not fully effective. Management of mining areas should also take into account adjacent ecosystems, which might be an important factor contributing to reducing the spread of contamination further into the environment. It is essential that forest management of such areas utilizes species that are tolerant to contamination and at the same time will

contribute to the contamination spread reduction. The primary function of these forests should be protective, since their stabilizing and erosion-reducing functions can effectively prevent the spreading of waterborne pollution to more distant areas when suitable species are applied.

**Author Contributions:** Conceptualization, V.P. and D.J.; data curation, T.B.; formal analysis, R.K.; funding acquisition, V.P., J.K. and M.B.; investigation, D.J.; methodology, D.J.; project administration, V.P.; resources, M.B.; software, D.J. and T.B.; validation, J.K., T.B. and M.B.; visualization, D.J.; writing—original draft, V.P., D.J. and T.B.; writing—review & editing, J.K., R.K. and M.B. All authors have read and agreed to the published version of the manuscript.

**Funding:** This research was funded by Ministry of Education, Youth and Sports of the Czech Republic, grant number FCH-S-20-6446.

**Conflicts of Interest:** The authors declare no conflict of interest.

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
