# Peer review of "The Need to Improve Riparian Forests Management in Uranium Mining Areas Based on Assessment of Heavy Metal and Uranium Contamination"

_forests, doi:10.3390/f11090952_

Round 1

Reviewer 1 Report

Revisions to forests-890988

Title: The Need to Improve Riparian Forests Management in Uranium Mining Areas Based on Assessment of Heavy Metal and Uranium Contamination

The work presented here is interesting, falls within the scope of the journal, and is well worth investigating. In the present study, the authors investigated how the exposer riparian forest to uranium and other heavy metals affects plant growth, assessing their phytoremediation potential. Leaves were collected at specific points. There are some concerns with the study, regarding soil sampling. In fact, no soil around the tree saplings was collected nor the level of uranium and heavy metals quantified in the surrounding area. Material and methods sections are described adequately. The manuscript is generally well written. The discussion is comprehensive.

Major Remarks

Why were tree leaves only analyzed in the study? To establish a proper phytoremediation strategy, soil and root accumulation of heavy metals and uranium needs also to be tacked.

Minor remarks

Abstract. The aim of the study on the abstract is not clear.

Tables 1 and 2. Add standard deviations.

Please add significances to Figures, even if they are not significant.

Figure 4. Redraw Figure 4, the axes are not clear

Author Response

Thank you for reviewing the manuscript and your useful and factual comments. We have modified the article based on your recommendations. A detailed statement on your partial comments is as follows:

Major Remarks

  1. Why were tree leaves only analyzed in the study? To establish a proper phytoremediation strategy, soil and root accumulation of heavy metals and uranium needs also to be tacked.

The reason is that (1) the contamination status of the surrounding environment has already been historically documented or is monitored regularly and (2) tree leaves can serve as a better tool for larger-scale risk assessment.

(1) The situation about contamination status of the soil (and water) is described in the chapter “2.1 Study area”, where many studies documenting situation of the soil or water in the area are mentioned. That's why we did not consider it necessary to repeat the soil survey. However, the studies often do not take into account real mobility and bioavailability of the potentially toxic elements. Tree leaves, on the other hand, can serve very well to understand and evaluate the real situation.

(2) There are numerous studies dealing with the contamination status of the soil in mining areas worldwide, however, they often do not consider large-scale mobility and bioavailability of the potentially toxic elements. These are better demonstrated by the leaves. Soil or root contamination is not so easily accessible for other organisms and mobile as that in leaves.

The role of the forest as a complex is often overlooked in contamination studies. The intention of this study was to point out this fact as well and not to accelerate the trend focused only on soil as a common medium for pollution assessment.

Minor remarks

  1. Abstract. The aim of the study on the abstract is not clear.

The passage with the aim of the study has been modified. The current form is as follows: “The aim of this study is to evaluate the contamination levels of riparian forest stands and their interaction with pollutants on the example of two localities with long and short-term uranium mining closure.” We hope that this small modification is sufficient.

  1. Tables 1 and 2. Add standard deviations.

It is possible to add a standard deviation values to Table 1. However, the table is then too large, and it is necessary to split into two tables (see attached file). In our opinion, this extension of the article and the fragmentation of the text is an unnecessarily large intervention, which is why we left the original format. However, if you deem it necessary, we can insert the newly created tables into the article appendices. In the case of Table 2, it is not standard to add standard deviation values. In our opinion, this will not bring significant benefits to the reader, and on the contrary, it will greatly enlarge and obscure the table, which we would like to avoid. We therefore apologize, but this recommendation has not been accepted.

  1. Please add significances to Figures, even if they are not significant.

The required significances have been added to the figures.

  1. Figure 4. Redraw Figure 4, the axes are not clear

Since it was not possible to redesign/redraw the figure to increase its quality and solve the problem, the figure has been replaced by a table.

Reviewer 2 Report

Comments and suggestions for the authors:
A study entitled “The Need to Improve Riparian Forests Management in Uranium Mining Areas Based on Assessment of Heavy Metal and Uranium Contamination” was submitted to forest for publication. Manuscript writing, data interpretation in this article could have been better compared to this present format. Anyway, incorporated changes, comments, suggestions or opinions concerning the manuscript are as follows:
Major comments:
1. Language should be revised through the manuscript by Native speaker with similar field of experience.
2. “Results and discussion” sections were larger than necessary
3. In case of Figures 2 and 3: The author should amphasized on results and conclusions that aggre/ disagree with other(s), along with unnecessary descriptions and citations should be avoided.

Minor comments:
1. Lines 29-32, page 1: should be revised
2. Lines 73-76, page 2: should be rephrased
3. Lines 99-113, page 3: should be concised with minimum citation
4. Lines 122-129, page 4: Should be revised using past tense
5. Line 135, pahe 4: What does author mean by “individuals” should be clear
6. Lines 146-154, page 4: Larger than necessary, should be revised with minimum citations
7. Line 158, page 4: should be inserted one appropriate reference by mentioning acid digestion rations of chemicals (HNO3: H2O2)
8. Line 171, page 5: Author should be clear about “blank” component clearly
9. Line 176,page 5: subheading 2.5, “ Data treatment” should be replace by “Data analysis”
10. Lines 218-219, page 6: Does the author Indicated,” When focusing on heavy metals, the content of Cd, Pb and Zn in the leaves crossed that of these acceptable levels”?Please check and revised it.
11. Lines 230-231,page 6: Actually it is difficult to say “ a similar trend of accumulation is evedent” for a mass of tabulated value except graphical presentation. So please revised it with appropriate phrase.
12. Lines 248-301, pages 7-8: This section should be concised
13. Lines 356, page 10: “For example” should be added before “Faměra et al.”
14. Page 11, Figure 4, “level of correlation matrix”is very small in front, should be revised it.
15. Lines 435-475, pages 12-13: Should be concised, be removed several gaps among the small fragment of paragraphs. Actually, Conclusion section should be presented as one paragraph.

Author Response

Thank you for reviewing the manuscript and your useful and factual comments. We have modified the article based on your recommendations. A detailed statement on your partial comments is as follows:

Major comments:

  1. Language should be revised through the manuscript by Native speaker with similar field of experience.

The language has been revised based on the recommendation.

  1. “Results and discussion” sections were larger than necessary

The text in the section has been reduced based on the recommendation.

  1. In case of Figures 2 and 3: The author should amphasized on results and conclusions that aggre/ disagree with other(s), along with unnecessary descriptions and citations should be avoided.

The text relating to the Figures 2 and 3 has been significantly reduced based on the recommendation and unnecessary descriptions have been deleted.

Minor comments:

  1. Lines 29-32, page 1: should be revised

The text in that section has been revised and reduced based on the recommendation.

  1. Lines 73-76, page 2: should be rephrased

The passage has been rephrased for easier understanding.

  1. Lines 99-113, page 3: should be concised with minimum citation

The passage was revised and reduced based on the recommendation. As it is necessary to point out that other components of the environment (soil, water) of both sites are polluted on the basis of previous research (studies) and their research may therefore not be part of this study, the number of citations cannot be reduced.

  1. Lines 122-129, page 4: Should be revised using past tense

The passage has been rewritten using past tense.

  1. Line 135, pahe 4: What does author mean by “individuals” should be clear

A word “tree” has been added for easier understanding.

  1. Lines 146-154, page 4: Larger than necessary, should be revised with minimum citations

The passage was revised and reduced based on the recommendation.

  1. Line 158, page 4: should be inserted one appropriate reference by mentioning acid digestion rations of chemicals (HNO3: H2O2)

To clarify the digestion methodology, source references have been added, and the text has been extended to the following form: “Digestion of samples was accomplished according to EPA methods: EPA 3050B and EPA 200.8. To accomplish complete decomposition, the samples were boiled in 15 ml of nitric acid p.p. (65%) with continuous additions of hydrogen peroxide (3 ml), filtered, filter boiled with the addition of 10 ml of aqua regia (3 HCl : 1 HNO3), filtered, filter boiled again with the addition of 10 ml of aqua regia (3 HCl : 1 HNO3) and filtered. All 3 solutions were mixed and topped up to 100 ml with demineralized water.”

  1. Line 171, page 5: Author should be clear about “blank” component clearly

The following explanation describing the blank has been added to the text: “Blanks were prepared by identical procedure with identical amounts of acids and hydrogen peroxide, only without the sample.”

  1. Line 176,page 5: subheading 2.5, “ Data treatment” should be replace by “Data analysis”

The replacement has been done.

  1. Lines 218-219, page 6: Does the author Indicated,” When focusing on heavy metals, the content of Cd, Pb and Zn in the leaves crossed that of these acceptable levels”?Please check and revised it.

No, he did not. There was a small mistake and the sentence has been rewritten for easier understanding as follows: “When attention is aimed at heavy metals, it is evident that the content of Cd and Pb in the leaves at the site 1 did not exceed the lower toxicity limit in any case.”

  1. Lines 230-231,page 6: Actually it is difficult to say “ a similar trend of accumulation is evedent” for a mass of tabulated value except graphical presentation. So please revised it with appropriate phrase.

This sentence is based on the graphical output (Figure 2) as mentioned in the text in the same sentence so the trend is visible. However, we agree that it may not be evident. After the revision, the word “evident” has been replaced by the word “visible”.

  1. Lines 248-301, pages 7-8: This section should be concised

The text in that section has been reduced based on the recommendation.

  1. Lines 356, page 10: “For example” should be added before “Faměra et al.”

The text has been modified based on the recommendation.

  1. Page 11, Figure 4, “level of correlation matrix”is very small in front, should be revised it.

Since it was not possible to effectively redesign/redraw the figure to increase its quality and solve the problem, the figure has been replaced by a table.

  1. Lines 435-475, pages 12-13: Should be concised, be removed several gaps among the small fragment of paragraphs. Actually, Conclusion section should be presented as one paragraph.

We agree that conclusion section should be consistent and presented as one paragraph if it is possible. However, in our opinion, the gaps are needed for better fluency of reading and a better orientation of the reader in this case of such length conclusion. There are many articles in the journal with such a formatted conclusion section (often much shorter). Therefore, we apologize but this recommendation has not been accepted.

Round 2

Reviewer 2 Report

Comment for authors:

MS: forests-890988

Incorporated changes made this manuscript better than before. Minor spell check and grammatical corrections required through the manuscript before make it final, as plenty of corrections have been performed with track change mode in word file.

Author Response

Dear reviewer,

thank you again for reviewing the manuscript and your useful and factual comments. Minor spell check and grammatical corrections have been done through the manuscript as recommended. The grammatical corrections were made by an expert in the field with a C2 Proficiency qualification. Based on that, we hope that everything is grammatically in order.

Sincerely

David Juřička
